# Understanding the Intrinsic and Extrinsic Motivations Associated with Community Gardening to Improve Environmental Public Health Prevention and Intervention

**DOI:** 10.3390/ijerph16030494

**Published:** 2019-02-11

**Authors:** Monica D. Ramirez-Andreotta, Abigail Tapper, Diamond Clough, Jennifer S. Carrera, Shana Sandhaus

**Affiliations:** 1Department of Soil, Water and Environmental Science, University of Arizona, Tucson, AZ 85721, USA; ssandhaus@email.arizona.edu; 2Department of Community, Environment and Policy Department, Mel and Enid Zuckerman College of Public Health, University of Arizona, Tucson, AZ 85724, USA; 3Boston Medical Center, Boston, MA 02118, USA; abbytap@gmail.com; 4New York Medical College, School of Medicine, Valhalla, NY 10595, USA; clough.d@husky.neu.edu; 5Department of Sociology and Environmental Science & Policy Program, Michigan State University, East Lansing, MI 48824, USA; jcarrera@msu.edu

**Keywords:** urban gardening, intrinsic motivations, extrinsic motivations, environmental health, soil quality, exposure assessments

## Abstract

Considering that community members continue to garden in and near environments impacted by pollutants known to negatively impact human health, this paper seeks to characterize the intrinsic and extrinsic motivations of a gardener and elucidate their perception of soil quality and environmental responsibility, awareness of past land use, and gardening behavior. Via semi-structured interviews with community gardeners in the Boston area (*N* = 17), multifactorial motivations associated with gardening as well as ongoing environmental health challenges were reported. Gardeners are knowledgeable about their garden’s historical past and are concerned with soil quality, theft, trash maintenance, animal waste, and loss of produce from foraging animals. Study findings directly inform the field of environmental health exposure assessments by reporting gardening duration, activities that can lead to incidental soil ingestion, and consumption patterns of locally grown produce. This information combined with an understanding of a gardener’s intrinsic and extrinsic motivations can be used to develop urban agricultural infrastructure and management strategies, educational programming, and place-based environmental public health interventions.

## 1. Introduction

Participation in community gardens provides many advantages—from reducing the cost of foods [1,2] to improved cardiovascular health [3,4,5] and mental health. [6,7]. These motivations can contribute to an improved health profile among urban gardeners, and community gardens can be viewed as a place-based strategy to address public health challenges, such as health promotion and environmental exposures. Small scale and community-based gardens are re-emerging as ways to improve food quality and transform industrial food growing practices. Gardens are an “example of a community-based environmental change that transcends age, ethnicity, race, income, and education, and thus provides an important example of a place-based strategy that can strengthen and sustain neighborhoods and improve residential health across the lifespan” [7].

Food gardening is on the rise and individuals are participating in gardening to, for example, have better tasting food (58%), save money (54%), grow food of higher quality (51%), spend time outdoors (35%), teach kids to garden (25%), and live locally (21%) [8]. In 2013, 37 million American households participated in food gardening at home, three million grew food at a community garden, and 76% of the households grew vegetables [8]. Unfortunately, gardening is occurring in and near environmentally compromised areas with contaminants known to negatively impact human health [9,10]. In particular, urban garden soils have been impacted by active and legacy sources of arsenic, heavy metals, asbestos, and organic compounds, as well as polycyclic aromatic hydrocarbons [11]. In some gardens in Roxbury and Dorchester, MA soil analyses have revealed high concentrations of lead in gardens, where average lead concentration measured in 141 gardens was 950 milligrams per kilogram (mg/kg), highlighting the city’s long history of lead-based paint, industrial activity, incinerators, and motorized vehicle traffic [9]. Few gardeners are testing for pollutants, and uncertainties about responsibility and specific cleanup standards are recognized as barriers to implementing locally based agricultural projects [10]. To not discourage urban agricultural activities, efforts are needed to assess a gardener’s motivation and behavior, which can then inform environmental risk assessments and urban agricultural policy and management.

To better understand the motivation of gardeners, Self-Determination Theory is applied. SDT is a broad framework used to understand individual motivation, which includes (among others forms) intrinsic and extrinsic forms of motivation as well as how social and cultural factors can support or undermine motivation [12]. Ryan and Deci (2000) stated that when conditions support an individual’s basic psychological needs—such as competence, relatedness, and autonomy—they will have increases in overall motivation, which impacts performance.

Intrinsic motivation, in general, is defined as being motivated for inherent satisfaction, interest, and enjoyment. In contrast, extrinsic motivation is defined as being motivated for the instrumental value or doing something because it leads to a separable outcome [13]. For example, community enhancement and relationship building are a form of “relatedness” and considered extrinsically motivating factors [13]. Relatedness pertains to the development and maintenance of relationships, belonging to a group, and/or being motivated to participate in an activity because it is valued by significant others to whom an individual feels (or would like to feel) connected [13].

To characterize the social and cultural dimensions of gardening, SDT is applied and participant’s responses are catalogued and assessed as they relate to intrinsic and extrinsic motivations. In parallel, a gardener’s perception of soil quality and environmental responsibility, awareness of past land use, and gardening behavior are also captured to inform environmental health exposure assessments. It is hypothesized that by gathering local-based information, we can use this knowledge to develop an infrastructure that supports urban community gardening, improve exposure assessments, and propose place-based public health interventions that are rooted in motivational research.

## 2. Materials and Methods

Semi-structured interviews were conducted with urban community gardeners who grow ornamentals and/or food at a garden managed by the Boston Natural Areas Network (BNAN). BNAN is now part of The Trustees of Reservations, a Massachusetts-based non-profit organization working to protect “special places” across Massachusetts with an emphasis on land preservation and sustainability [14]. The study protocol for research was approved by the Northeastern University’s Human Subjects Research Protection Program (IRB#: 13-10-03).

### 2.1. Recruitment

Recruitment and interviews took place between March and October 2014. Participants were recruited through a variety of outreach methods. At the BNAN annual gardener’s gathering in March 2014 and a BNAN event at City Soils in Mattapan, researchers explained what the study involved and how they could participate. People who signed up had the option of being interviewed that day or leaving their information to be followed up with later. Additionally, participants were recruited at community gardens or by word of mouth. There were no restrictions based on residential area or amount of time spent gardening, and participants had to be over 18.

### 2.2. Interviews

Interviews were conducted either in person at a garden or a home, or over the phone. The interviewer followed a semi-structured script (see Appendix A) and each interview was recorded with two recorders for backup purposes. At the close of each interview, the interviewer asked a set of questions to gather demographic information. For their participation, participants were given a gift bag that included a hand trowel, gardening gloves, and gardening twine.

The interview questions began with how the participants came to gardening, the time they spend gardening, and what they had learned so far. They were asked about their routines, what they grow, and their methods of growing, as well as concerns they have about their gardens and their environment. Finally, they were asked about how they feel when they garden, and the benefits of gardening (see Appendix A). The interviews lasted between 14 and 58 min. The average interview lasted 31 min.

### 2.3. Coding

All interviews were recorded, de-identified, transcribed, and coded. All sessions were transcribed via VerbalInk, a transcriptions service company (VerbalInk, Los Angeles, CA, USA, 2017). An initial coding scheme was created that closely resembled the interview questionnaire. Then, transcripts were divided among four researchers to begin thematic coding. Next, researchers informally compared coding results to assess interrelated reliability and discussed the emergence of new themes that were not captured in the initial coding scheme. The codebook was revised and finalized through an iterative group process. In some cases, this involved creating new and/or merging categories. Differences between the team members’ coding results and the addition of new codes were discussed and reconciled to prevent inappropriate coding. Once this codebook was finalized, all interviews were systematically coded using NVivo 11 (QSR International, Melbourne Australia). A total of 12 major/parent codes were identified, each with their own set of specific minor/child notes: Motivation–Community, Motivation–Individual, Concerns, Demographics, Exposure, Formal Learning, Garden Organization, Informal/Free-Choice Learning, Knowledge of Historical Land Use, Learning to Garden, Protection, and Responsibility. Two coders were assigned to analyze each interview and met periodically to compare coding results and ensure appropriate coding. The results were then categorized by whether the recorded observation was motivated intrinsically or extrinsically (see Section 1 Introduction for more detail).

## 3. Results

Below is an outline of the major results of the study. Sections are divided by the seven following categories: gardener demographics, intrinsic motivation (i.e., inherent satisfaction, interest, and enjoyment), extrinsic motivation (i.e., instrumental value or doing something because it leads to a separable outcome), recommended governance for environmental health safety, concerns in their garden and neighborhood, awareness and understanding of historical land use, and exposure assessment - activities in the garden and consumption patterns. Within these categories, authors highlighted the major findings that arose from interviews that would inform urban agricultural infrastructure and management strategies, educational programming, and place-based environmental public health assessments and interventions.

### 3.1. Gardener Demographics

Of the seventeen participants, the majority were female (*n* = 13), with two male participants, one unlisted, and another who listed their gender as male-bodied/complex. Eleven were white, and six participants did not specify their exact ethnicity. Two participants gave ranges for their age and five were unlisted. The average age of the remaining participants (*n* = 10) was 47.8 years old.

### 3.2. Intrinsic Motivation

#### 3.2.1. Mental Well-Being

All participants (*N* = 17) indicated that they received some psychological benefit or reward from gardening when answering one or more of the following open-ended questions: “What led you to gardening? Why do you garden? What motivation have you received from gardening? What are your thoughts regarding the health motivation from gardening?” Responses ranged from general positive feelings to therapeutic qualities, such as alleviating stress. Of the participants, 47% described gardening as relaxing, and 59% talked about how being outside (even just for a few minutes each day) gave them a much-needed break. Four participants experienced feelings of spirituality or closeness to god in their participation in gardening. Of the 47% of participants who stated gardening was relaxing, five participants cited gardening as being relaxing for a specific reason. For example, a participant stated:
I find it very relaxing. It’s hard work when you start it. You don’t realize how hard it is, but it’s rewarding and relaxing at the same time… It’s good mental health, good physical health, you get a lot of fresh air outdoors. We get good food. We know what we’re eating.

One participant said, “I think it’s—actually I think taking out the weeds, I feel like there’s something nice about just touching the earth. It’s kind of like a different kind of yoga. It’s very calming. It’s nice to work with the earth.” Patience and peacefulness were cited as well. One participant stated:
As a gardener, you’re [not] making anybody do anything. You can’t force things. The best thing you can do is deeply understand what things are going to do naturally, because everything is going to act in its own best interest whether it’s a plant or a person or a cat. What you—if you want certain outcomes the best way to achieve that is to create conditions into which something will flourish in a controlled closed certain way. You want to create conditions under which something will flourish but in a direction that you may be able to manipulate.

#### 3.2.2. Intergenerational Learning

Another intrinsic form of motivation that stood out in the dataset was intergenerational learning. Intergenerational learning is a factor in social contexts that can produce variability in intrinsic motivation; in this case, the want to garden or not. Intergenerational learning is defined as knowledge passed from one generation, often parents or grandparents, down to the person in question, but it also includes younger generations sharing knowledge and skills with older generations. Five participants talked about learning from their grandparents. One gardener stated, “It’s something that’s passed down. My mother loved gardening and always had a garden. My grandmother had an incredible green thumb…” While another participant highlighted several elders playing a role in their learning:
Because of my grandmothers. Actually, I have two grandmothers and an adopted elderly woman who lived in my neighborhood, who acted as my grandmother, and all three of them are really into working with plants, and so I learned to love a garden from them.

### 3.3. Extrinsic Motivation

In addition to the many intrinsic motivations experienced by the participants in this study, there were also some very important extrinsic motivations of urban community gardening that were frequently expressed. Other similar studies have observed that ecological considerations and concerns, a sense of responsibility, and a sense of an enhanced community are associated with community gardening. Similarly, all the participants in our study (*N* = 17) expressed these positive extrinsic motivations.

#### 3.3.1. Economics

One extrinsic theme that emerged during participant interviews was the economic motivation of urban community gardening. Many gardeners mentioned, specifically from a financial standpoint, how getting food from their garden is much less expensive than buying the same amount of harvested produce from a grocery store. This theme focuses on how participants described the gardening as a benefit in which they are able to save money on produce. Some participants described themselves as being on a fixed income. One participant specifically stated that they “Usually try to plant what’s most expensive from the stores, like tomatoes [and] sugar snap peas.”

In addition to the benefit of reducing the cost of the produce grown in gardens, which may otherwise be purchased for a higher price in grocery stores, participants expressed satisfaction about the quality of the produce. One participant stated that getting involved in community gardening “…Brought down [their] grocery bills in the August [and] September months” and went on to say that “There are times when I’ve taken about $40.00 worth of vegetables from my garden.”

Regarding the quality of produce, participants expressed a satisfaction of knowing what they put into their food and being able to reap what they sow. A participant stated that “it makes you feel good; you appreciate food [and] vegetables; you appreciate what it is, the task behind it.” Others highlighted accessibility of produce, stating, “I can pick one [vegetable] and then an hour later I can be having my dinner.”

#### 3.3.2. Community Enhancement and Building Relationships

Four participants indicated that they gained friendships and other social relationships through their participation in community gardening. These relationships can be parlayed into many other factors of these participants’ lives and foster relatedness ([13] see Section 1 Introduction for definition). One participant even said, “There’s a woman who became my mentor for the next several years, and taught me about gardening and got me a job gardening, and [began] my whole life.” Twelve participants stated that their gardening experience and knowledge was influenced in part by their relationships with various family members who had encouraged them to garden. A participant’s sentiment:
It’s a great networking vehicle for meeting, and working with, and getting to know people that—who— even though we live in the same neighborhood, paths may not cross because we have different backgrounds and different involvements, but there’s this commonality in gardening that brings people together, and it’s really great.

Participants (*n* = 5, 29%) also reported learning different techniques that were unique to their area from their neighbors. One gardener stated they “…talked to all the Italian neighbors around me that used to have gardens and got their advice.” Below is a thought from a participant capturing their motivation for communal relationships:
When you’re gardening at home it’s one thing, but when you’re gardening with other people that are doing stuff that’s interesting, you can say how did you do that? Or why are mine dying and yours living? It’s…really great…So you learn a lot from people definitely.

Another important extrinsic motivation widely mentioned by study participants was the benefit of coming together, cooperating, and sharing within a community. Some mentioned the pleasure of being able to take part in garden events and enjoy the community. Gardeners expressed how they are making new friends, partaking in new activities, and that the garden gives them a voice in the community. One participant stated, “Without community space, without space that’s public and held in common then it’s really hard to have a consistent community voice.”

In addition to coming together as a community, sharing garden produce was also an extrinsic form of motivation among urban community gardeners. Many of the gardeners said that they share what they have grown in the garden with friends, family, and neighbors. One participant stated, “Well, I think there’s a benefit in just being social with people, that it’s nice to be able to give… to pass vegetables along.” The gardeners emphasized that they were willing and happy to share when they had an abundance of produce.

Overall, participants stated that it is rewarding to feel like they are part of a community, meet others that share a common interest, share produce and knowledge, and be able to save money and control the quality of your produce. As one participant stated, “Everybody can find common ground in a garden.”

### 3.4. Recommended Governance for Environmental Safety

#### Gardener’s Safety—Who is Responsible for the Safety of the Soil and the Vegetables that are Being Grown in the Garden?

Ten participants (58%), with three being home gardeners, agreed that they are responsible for the safety of the soil and vegetables being grown in their garden, particularly at their individual community garden plot. However, there were times (*n* = 3) when participants said that more than one entity was responsible. One stated that all the users of the garden should work together to keep it safe, while another said that individuals are responsible for their own gardens but must abide by the rules of their organization. Another participant made the distinction that the gardener is responsible for the care of their vegetables grown, but that if the soil is found to have contamination, then the city government is responsible. They further explained that they are not comfortable growing something in their backyard, but “I’m comfortable growing in here [community garden] because I think the city’s done something, done all the testing required.” Two other participants stated that that the government and organizations were responsible, with one stating that the “coordinators oversee [the garden] with the help of the BNA (Boston Natural Areas Network).” Those who listed multiple groups or persons responsible or combined personal responsibility with another group listed the Boston Natural Areas Network (which owns many of the community gardens), the city itself, or the Massachusetts Department of Conservation. Examples of reported mutual responsibility by participants include:
… the community needs to assess—get levels of contamination and make a plan for themselves. I think basically it is the responsibility of neighbors, residents, gardeners, and that there should be support/funding available from government.
Interviewer: *Who is responsible for the safety of the soil and the vegetables that are being grown in the garden?*
Interviewee: *The people who eat it I guess.*
Interviewer: *Yeah?*
Interviewee: *I would tend to think so. I don’t care to rely on government for much. I would say that people growing in soil, especially in a space where there’s a coordinator or whatever that it would be pretty easy for people to if they want the soil tested to test it themselves. It can be quite expensive if you have a large piece of land, and so there should be public funds available for it I think.*

### 3.5. Concerns in their Garden and Neighborhood

When participants were asked, “Are there any environmental or health issues neighboring your garden or in your neighborhood?”, responses varied. Major themes that arose were issues related to soil and food safety, animals, stolen food, and trash.

#### 3.5.1. Soil Quality

Many participants expressed concern over the quality of their grown produce, with quite a few expressing concerns of lead in the soil. One participant was particularly concerned, stating that their garden “is on the side of the driveway and there’s a car there all the time” and that they “can’t imagine that it doesn’t have lead in it”. Another was concerned that their neighbor may be spraying their garden with materials that may drift over to their own, and one was concerned about the quality of the compost used in their garden.

Thirty-five percent of participants stated that they had done soil testing on their gardens, and an additional 35% expressed interest in doing testing. Only one said that they had done water testing, and two more expressed interest in it. The participants who had not done soil testing but wished to stated that they would have done so if it were not for the monetary cost of the tests. One participant, however, stated that while they test their own soil, others may not do so because they would rather not know. The participant went on to say that people may not be able to afford to treat their soil if it is contaminated, and that people may prefer to continue to garden in their soil and simply avoid the issue of addressing contamination entirely.

#### 3.5.2. Store-Bought Food Quality

Some gardeners chose to grow their own vegetables because they have concerns about store bought produce. Five participants (29%) stated that they were concerned about pesticides in store-bought produce. One stated that when they bought conventionally grown fruits and vegetables, they diligently washed the produce to remove pesticides.

Four participants (23%) were concerned about “germs” and handling of store-bought food. Some respondents spoke of how produce at grocery stores can be handled by anyone visiting the store, while one participant said that they were concerned “about the types of bacteria that might travel along with those vegetables wherever they came from”. This concern also ties into gardeners concerns about travel distance of store-bought produce, which three gardeners (17%) discussed. In addition to bacteria in the shipping process, one participant was concerned about the health of the workers who grow and transport the produce. One preferred garden-grown produce because the time from garden to table is much shorter.

Three participants (17%) discussed genetically modified organisms (GMOs). One participant stated their belief that produce in grocery stores is bred for longevity and that the flavor is bred out, while another was concerned about purchasing GMOs from grocery stores without knowing it.

#### 3.5.3. Theft and Waste

Four participants (23.5%) stated that theft was a problem in the community gardens. Of those, one participant had experienced theft and described how people jumped the fence to steal vegetables. Another noted that they had to lock the garden to protect it from theft. One interviewee stated that the garden was broken into and while vegetables were not stolen, trash was left.

Another notable concern was trash and waste (*n* = 5, 29%) in the garden. Three participants stated that their gardens did not have regular trash pick-up, which makes it difficult to get rid of waste. The gardeners have to collect their own trash and take it home or to a pick-up location to dispose of it, and they say that it is a “hassle” or “hard to get rid of”. In addition to trash, two participants discussed animal waste as an issue. Participants expressed interest in trash pickup in the gardens. One participant stated that there used to be trash pickup at their community garden, but now those who use the area have to bring their own gardening waste back to their houses, which can be time consuming and exhausting. Of the participants who gave their ages (*n* = 10), the average age was 47.8 years old, and for those on the elderly side or for those who may have any sort of disability or mobility restrictions, transporting their trash may be difficult or dissuade them from gardening in the first place. Having a designated location for trash and a regular trash pickup schedule could help to alleviate concerns and increase the ease of waste disposal.

One said that cat feces were a “big problem in the garden”, while another was concerned about fecal matter from other animals, such as raccoons, squirrels, or rabbits. In addition to animal waste, damage from animals was a concern of the gardeners. Three participants (17%) spoke of animals and rodents in the garden. All three discussed rodents coming in and eating their vegetables, with two discussing a specific rabbit. One interviewee put up a fence to keep the rabbit out and closed a hole that the rabbit could get through to protect their produce.

### 3.6. Awareness and Understanding of Historical Land Use

Thirteen (76%) of the participants were able to discuss historical use of their garden areas. Many (*n* = 11) could pinpoint exactly what the area was before they began using it. In some cases (*n* = 5), participants reported that their garden areas were always gardens or used for agriculture, and in others, it was the yard of a house or an industrial lot. One participant knew what the land was used for one hundred years ago, while the others did not stretch quite as far back. In general, those who were able to discuss the historical land use framed it in the context of potential hazards. For those who had gardens in spaces previously used for industry or [conventional] agriculture (*n* = 7), there was fear of contamination. Another participant stated that their garden was previously a used car lot, stating it was a “particularly bad” use.

### 3.7. Exposure Assessment—Activities in the Garden and Consumption Patterns

In order to inform human health exposure assessments, questions regarding gardening frequency and behavior were posed. Participants stated their gardening frequency in two different ways—hours per week and number of times that they garden per week. For those who stated their frequency in hours per week, nine participants stated they garden for 0–5 h, followed by four participants that garden for 5–10 h/week. For those who gave their frequency in number of times they visit the garden per week, the most common response was that they visited the garden 1–3 times per week. In addition, this varied depending on the time of year and weather. One participant stated that they would skip going if they knew it was going to rain, while many said that they spend more time in spring when it is planting season.

When at the garden, participants are most commonly watering, planting, weeding, spreading amendments, and turning soil (Figure 1). For those participants who responded, 76% of participants wash their hands after gardening and all participants reported washing their produce after harvesting (Figure 2).

When describing how much of their garden they eat, one participant said, “We eat pretty much all of what we grow. It’s a small garden,” and another said, “I eat everything that grows in there. I don’t let it go to waste.” When describing how much of what they eat comes from their garden, one participant said, “Sometimes all my meals are just from my garden in peak harvest time,” while another participant said, “Less than 5% of my total calories come from my home garden.” Many of the gardeners eat everything they are able to grow from their garden.

### 3.8. Limitations

The main limitation in this research was that the pool of participants was mostly based around BNAN community gardens. This potentially left out gardeners who were not affiliated with BNAN. Participant recruitment and availability was limited, and on occasion, people would sign up to be interviewed but would not be able to complete the interview due to their time constraints or those of the interviewers. Additionally, different demographics were not actively sought out. This could lead to bias in the study results based on gender, age, race, and place of residence. Lastly, when asking participants how much they eat from their garden, participants answered with a general response that did not allow for an exact calculation of vegetables consumed from their gardens.

## 4. Discussion

By applying the Self-Determination Theory, we gained a deeper understanding of what intrinsically and extrinsically motivates individuals to garden, even when their environment has been compromised by pollution. These findings increased our understanding of a community gardener’s behavior, perception of soil quality, and awareness of past land use, which highlighted environmental health knowledge gaps and potential barriers to the sustainability of urban agricultural activities.

### 4.1. Intrinsic Motivation

Intrinsically motivating behaviors are those that are engaged in for their own sake or for the pleasure and satisfaction of performing them [12,13]. In this study, all participants (*N* = 17) indicated that they received some psychological benefit or reward from gardening. In Van de Berg and Custer’s 2011 study on stress and gardening, they found “positive mood was fully restored after gardening” [15]. Their findings are consistent with other studies that observed that increased contact with nature lowers mortality and morbidity from stress related diseases [16,17]. Even just being around microbes in the soil has been shown to cause an increase in serotonin; the bacteria, *mycobacterium vaccae*, activates a set of serotonin-releasing neurons in the brain [18].

In our study, 65% (*n* = 11, 65%) of participants reported interest in learning gardening techniques from either a family member or gardening colleague. Intergenerational learning is an important part of lifelong learning [19], where the generations work together to gain skills, values, and knowledge. Cognitive Evaluation Theory (CET), a sub-theory of SDT, may explain why intergenerational learning is motivating. CET attempts to account for the factors in social contexts that produce variability in intrinsic motivation and proposes that interpersonal events and structures like communication and feedback can increase an individual’s self-efficacy, which then enhances intrinsic motivation [13]. When applying CET in the context of gardening and inter-generational learning, individuals are intrinsically motivated because they are gaining confidence and satisfying their basic psychological need for achieving competence [13].

Beyond the transfer of knowledge, intergenerational learning fosters reciprocal learning relationships between different generations and helps to develop social capital and cohesion in our ageing societies [20]. Eleven of the participants cited their parents as being a source of knowledge in how to garden as well as an inspiration to continue gardening. Intergenerational learning has also been shown to contribute to a greater affinity for gardening and higher continuing levels of participation [21]. A study on intergenerational gardens across six U.S. cities observed that youth were motivated to be in the gardening program and most likely joined because of the influence of an adult, and conversely, adult gardeners were motivated by the opportunity to work with youth [21]. Furthermore, intergenerational learning among youth and grandparents plays a key role in the passing of cultural knowledge to younger generations. This is particularly the case among families that migrated from their countries or communities of origin, where they shared similar goals with elder family members. This knowledge exchange contributes to their personal goals [22]. Mayer-Smith et al. (2007) further expanded the notion of intergenerational learning and demonstrated that by combining urban farming with community elders, elementary students, and their teachers, the experience can foster an environmental consciousness and provide valuable informal environmental education [23]. Rahm’s 2002 program in Greeley, Colorado found a bevy of educational opportunities for youth through their participation in an urban gardening program [24].

Surprisingly, even though eleven of the participants cited their parents as a source of knowledge and motivation to garden, no participant reported learning about any historical past land uses or environmental contamination from a parent or other family member. This could be due to recent migration to the area or lack environmental health awareness. Since intergenerational learning is abundant in garden settings, this finding highlights the need for environmental health to enter the conversation and become part of family history. Intergenerational environmental health education could be a key to increasing overall environmental health awareness.

### 4.2. Extrisic Motivation

Participants discussed economics and saving money as motivations to garden. Gardening has been shown to reduce the amount that an individual or family spends on groceries by providing them with produce that they would otherwise have to purchase. Over 33% of gardeners in Newark, New Jersey reported that they gardened because it saved them money [1]. This is supported by another study where respondents reported that saving money was “highly valued”, especially among minority women [25].

Participants also discussed community-associated motivations with gardening. Gardening has been previously identified as a neighborhood-building activity—a way to reduce social barriers and improve neighborhoods [1]. Participants reported that they feel more connected to their community and neighborhood as a result of gardening, which serves as an extrinsic factor to sustain gardening activities and connect with others. Teig et al. (2009) observed numerous gardener motivations—from increased social connections and civic engagement, community building, and increased sense of belonging from their participation in the garden to being a “positive social influence” [7]. This finding further supports observations that gardens can create community-based environmental change and are a “place-based strategy that can strengthen and sustain neighborhoods and improve residential health across the lifespan” [7].

Organismic Integration Theory (OIT), another sub-theory of SDT that aims to identify the factors in social contexts that produce variability in extrinsic motivation [13], may explain these results. Under OIT, these reported extrinsic motivations directly correlate with providing an immediate reward (reduction in food costs, saving money), approval from others (meeting people with similar interests), and/or a seeing a conscious value in gardening. Additionally, community enhancement and relationship building is a form of “relatedness” [13], one of the conditions that supports an individual’s basic psychological needs, which can lead to increases in overall motivation and performance.

### 4.3. Study Implications for Environmental Health

While there are studies that explore the reported motivation of gardening, this study digs deeper into a gardener’s behavior, environmental health concerns, and what type of support a gardener needs to sustain urban agricultural endeavors like food gardening. In this study, it was observed that gardeners are aware of their surroundings (*n* = 13, 76%), as opposed to Kim et al. (2014), who observed that, in general, study informants were not aware of historical past land uses or what the land was used for prior to a community garden due to information barriers [26]. It is important for gardeners to understand the historical past uses of their garden areas to have a sense of whether their soils are safe. According to a 2013 U.S. General Accounting Office report, one in four Americans lives within three miles of at least one hazardous waste site [27]. Also, commercially available gardening amendments (e.g., fertilizers) may contain heavy metals and need to be investigated further [28]. This is especially relevant to the participants, as Heiger-Bernays et al. (2009) revealed that the compost provided to Boston community gardens contained elevated levels of polycyclic aromatic hydrocarbons (PAHs) and lead [11].

Not washing hands and produce can lead to incidental routes of soil ingestion [10]. The results reported here are reassuring and demonstrate that the participants are following recommended protective practices [10]. Currently, exposure assessment calculations that are conducted in communities neighboring hazardous waste sites estimate that the average adult indirectly ingests 10 milligrams of soil per day (mg/day) with an upper percentile of 50 mg/day [29], and when accounting for soil and dust, it is 30 mg/day with an upper percentile of 50 mg/day. If a community member gardens, this intake rate may increase.

In addition to incidental soil ingestion, public health officials also need to be aware of certain plants that can accumulate pollutants in their edible tissue. Studies have seen increases in healthy food consumption in association with gardening [30,31]. An adult who participated in a community garden consumed fruits and vegetables 1.4 more times per day than those who did not participate, and they were 3.5 times more likely to consume fruits and vegetables at least five times daily [31]. Because gardening is associated with increased consumption of fruits and vegetables [31], it is important to determine site-specific uptake patterns of pollutants into commonly grown crops and consumption patterns. In this study, participants reported growing a variety of crops ranging in plant families, with the most popularly grown crops being tomatoes, eggplant, lettuce, and herbs (unspecified) (Figure 3). Identifying the crops people grow and the amount of time gardeners are in the garden can directly inform exposure assessments. Data from urban and rural gardens reveal that, in general, when compared to the U.S. Food and Drug Administration’s Market Basket Study (i.e., what could be expected from a typical U.S. grocery store), the locally grown vegetables accumulated more harmful elements [32,33]. Produce from certain plant families (e.g., Asteraceae, Brassicaceae, and Apiaceae) accumulated more arsenic, lead, and cadmium than others, and although they themselves might not be the major source of exposure, they may contribute to an already chemically burdened body [32].

### 4.4. Who Is Responsible for Environmental Quality and What Support do Gardeners Need?

A goal of this research was to inform the management of community gardens and other urban agricultural spaces. Although participants had varying opinions on who should be responsible for the safety of their gardens, there was consensus on what type of support gardeners needed. Most gardeners (*n* = 10, 59%, three home gardeners) stated that they were responsible for the safety of the soil and vegetables being grown in their garden, particularly within their individual community garden plot. However, some participants made the distinction that the gardener is responsible for the care of their vegetables grown but that the city government is responsible for soil quality and should have public funds available for community members to get their soil tested.

No participant stated that they were not interested in soil or water testing. Many (*n* = 8) were interested but were not able to or did not wish to spend the money on the analyses. This outcome echoes observations from a study conducted in Baltimore in which gardeners reported the need for a local testing service or a government-funded public service for soil testing [26]. In contrast, key informants in Baltimore recommended that citywide interventions bypass the need for gardener knowledge altogether [26], whereas in this study, most participants reported they were aware and concerned but that they themselves were responsible for their soil and vegetable quality. Although it was observed in the aforementioned study that there are information barriers to conducting site history and soil tests, in this study, participants stated that they know to test, but only eight participants conducted any environmental analyses at their garden (seven completed soil analyses and one participant conducted a water analysis).

Soil testing is critical, especially in urban areas where children may play, garden, and consume produce. In 2013, a community garden space in Boston was remediated due to elevated soil lead concentrations ranging from 305 to 1643 mg/kg due to past land uses and housing built prior to 1979 likely painted with products containing lead [33,34]. For perspective, the current U.S. Environmental Protection Agency uses a residential regional soil screening level of 400 mg/kg [35]. When assessing lead exposures in children, indirect soil ingestion is especially critical and can lead to elevated blood lead levels that can cause irreversible neurological developmental issues [36]. According to the U.S. Center for Disease Control, there is no safe blood lead level in children, meaning that any exposure to lead can potentially lead to irreversible damage and affect a child’s performance in school, IQ, and attention span [36]. Based on this feedback and the potential sources of lead and other heavy metals in urban spaces, it is recommended that community gardeners be offered free or low-cost soil and water quality analyses to monitor for potential contaminants of concern.

## 5. Conclusions

Through semi-structured interviews with community gardeners in the Boston area (*N* = 17), authors contributed to the field of environmental health assessment while simultaneously disentangling the intrinsic and extrinsic motivational factors that keep people gardening, even when there are neighboring sources of potential contamination. This study illustrates how community gardens provide psychological motivation, feed extended family networks, and are spaces where community members learn from each other by sharing gardening tips to recipes. Each gardener takes their own approach to their plot, but it is also a shared space where individuals influence and advise each other. Participants also reported extrinsic motivations to garden, such as saving money, a reduction in food transportation efforts, and less worry about bacteria and GMOs. When it came to “knowing the land”, participants were knowledgeable about the historical past of their garden. Seventy percent of participants stated that they conducted or were interested in conducting an analysis of their garden soil and felt that they, along with local non-profit and governmental organizations, were responsible for the safety of the soil. In general, gardeners are mainly concerned with soil quality, theft, trash maintenance, animal waste, and loss of produce from animals. This study reveals the manifold intrinsic and extrinsic motivations associated with gardening and the ongoing environmental health challenges associated with urban gardening. Community gardeners are likely to consume more fruits and vegetables but need assistance and support to determine soil health and the presence of contaminants of concern. Community gardens have been shown to help address social and economic constraints on health by increasing access to wholesome foods, improving community building efforts, creating green space, and reducing the cost of foods. Home and community gardening are on the rise, and community revitalization efforts may be diminished if these gardens are located in environmentally compromised spaces. It is important for community members to have access to the necessary support, infrastructure, and soil/water/plant analyses to ensure they are gardening in a safe environment.

## Figures and Tables

**Figure 1 ijerph-16-00494-f001:**
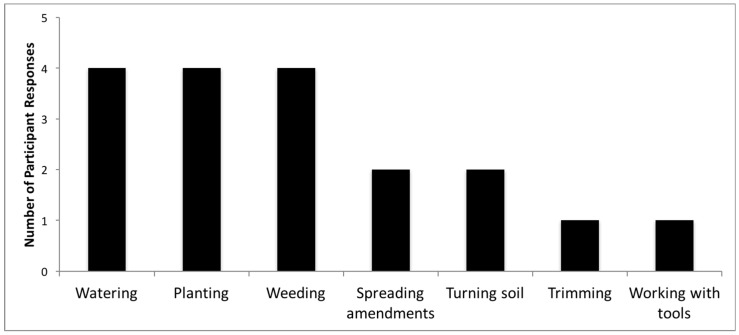
Participant activities in garden and contact with soil.

**Figure 2 ijerph-16-00494-f002:**
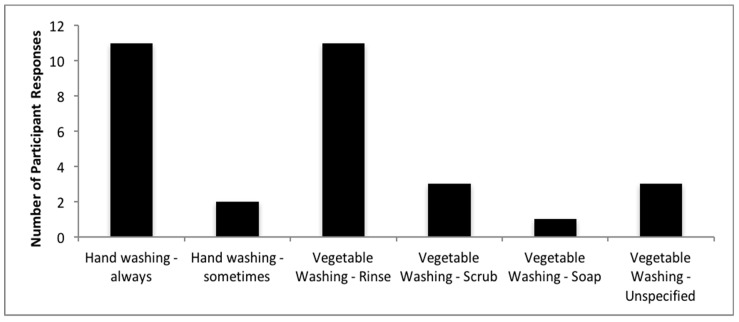
Hand and vegetable washing after gardening activities.

**Figure 3 ijerph-16-00494-f003:**
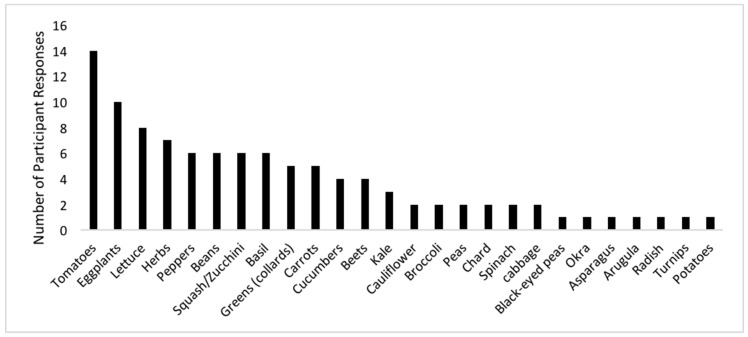
Crops typically grown by participants.

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
