# Peer review of "Understanding the Intrinsic and Extrinsic Motivations Associated with Community Gardening to Improve Environmental Public Health Prevention and Intervention"

_ijerph, 2019, doi:10.3390/ijerph16030494_

Author Response

RESPONSE TO REVIEWER 1 COMMENTS

January 23, 2019

Dear Reviewer 1,

We are resubmitting an original manuscript: “Understanding the Intrinsic and Extrinsic Benefits and Challenges Associated with Community Gardening to Improve Environmental Public Health Prevention and Intervention” (now with a slight title change) for publication in the International Journal of Environmental Research and Public Health. Thank you very much for the opportunity to revise and resubmit the manuscript.

We are grateful for the thoughtful comments we received from you regarding our manuscript. We appreciate the opportunity to revise the manuscript and submit it for your further consideration. We have taken into account all of your comments. Below are our responses to the comments.

Brief summary: The main goal of this paper is to explore self-reported intrinsic and extrinsic benefits of gardening, and environmental concerns regarding gardening—in particular, gardening that involves growing food. The authors seek to achieve these aims using semi-structured interviews with gardeners in the Boston area. The authors interviewed 17 gardeners and do a fine job of providing the reader with quotes from gardeners regarding the goals of the paper. However, ultimately, the paper falls short in overall organization, in providing novel insights to the field, and in reporting of methodology.

Broad Comments: While qualitative data provide critical insight into the effects of gardening on human health and well-being—meaning the bones of this article are sound and could be turned into a high-quality paper—the paper as-is feels unfocused and haphazard. Below I further outline two main strengths and two main weaknesses of the paper.

Author’s Response: Thank you for your thoughtful comments. By addressing the overall organization, selected components of the methodology, and further highlighting the novel insights to the field, this manuscript has been improved.

Strengths:

Data: The data reported in this article are rich and interesting. Given that the study of gardening and human health and well-being is relatively new, qualitative data can provide useful, guiding directions for future experimental studies of the causal mechanisms.

Environmental focus: The focus on environmentally-based health concerns is novel and adds to

the field. While there are many studies that explore the benefits of gardening that gardeners report, more research into gardener environmental concerns and awareness of environment dangers of gardening is needed—this paper provides such insight.

Author’s Response: Thank you.

Weaknesses

Organization: The overall organization of this article is somewhat haphazard and unfocused. For example, it is unclear what the gaps in previous literature are, and how this article is helping to fill those gaps—the paper needs a stronger, more upfront argument for its scientific contribution. Expanding and reorganizing the introduction would address this concern. Framing the benefits of gardening within Self- Determination Theory is somewhat novel and interesting, but ultimately this framework feels underdeveloped and unclear. This framework either needs to be explicitly threaded throughout the entire paper, or it should be dropped.

Author’s Response: We have added more information on Self-Determination Theory (Page 2, lines 12-23), and clearly state our scientific contribution (Page 2, line 24-30) in the Introduction section.

Also, a discussion section was added to further highlight the research’s scientific contribution  (Pages 9-12).

Organization and flow also feel weak due to the lack of clarity in the purpose of the data collection. The paper does not clearly outline why the data were collected the way they were, or what the ultimate purpose of collecting these data should be. The results section feels cobbled together rather than purposefully developed with a clear aim in mind. This, in part, led to my being unclear in the overall purpose of the paper. What are the “take-home” points, and how do these points fill gaps in the literature?

Author’s Response: We have added materials discussing the overall purpose of the paper (Page 2, line 24-30) and a section on the take-home” points, and how this research fills knowledge gaps in the literature (discussion section in general Pages 9-12, specifically Page 9, lines 16-21) and the conclusion (Page 12, lines 28-32).

Reporting of methodology: Overall, the methodology reported in the paper left me confused and unclear on how the authors coded and analyzed the data. More specificity is needed. For example, the paper reports that the transcripts were divided amongst four researchers, but that that two coders coded each transcript. Does this mean that each transcript was coded by a different pair of coders? How was the reliability of the two coders assessed? How many and what “codes” were included in the final codebook? How did these codes line up with the results subsections in the paper? More clarity here would be helpful. A final area where more clarity in coding schemes would be helpful is exemplified on page 3, where a participant is quoted for an example of how gardening requires patience; to me, however, this quote seems to drive home the degree to which gardeners have a sense of control. How did the authors decide that this quote exemplifies patience and not something else? Providing more detail on coding may help to clarify.

Author’s Response: More details regarding how the authors coded and analyzed the data was added to the method section  (Page 3, lines 16-31). We apologize for the confusion, the terms patience as well as peacefulness were used by interviewees and catalogued under “Benefit – Individual àPsychological.

Specific comments: Other organizational issues include reporting past research in the results section, which is typically a section free of past literature.

Author’s Response:  Past research results have been moved to the newly added discussion section of the manuscript (Pages 9-12).

Transparency/clarity in data: Occasionally, the data reported feel misleading. For example, on page 3 there is a sentence that reads “All interviewees…indicated that they received some psychological benefit or reward from gardening.” However, each gardener was prompted to report “the benefits of gardening” (p. 2). Given that the interviewers asked about benefits, participants may have stated benefits due to researcher demand. Another similar example includes a comment on page 8: “Twenty-nine percent of participants were concerned about trash in their gardens, so having a designated location for trash and regular trash pickup schedule would further help to alleviate concerns and increase the ease of waste disposal.” Here, “twenty-nine percent” feels overblown, as this is realistically only about 5 participants. It is unclear that this is truly a barrier to gardening and what the cost-benefit ratio of providing trash pickup would be. Perhaps interviewing nongardeners and asking why they do not garden would provide a better evaluation of barriers to gardening.

Author’s Response:  Thank you. We have added clarification to the beginning of the section: 3.2.1. Mental Well-Being. We also changed the wording “twenty-nine percent” to “Five participants and modified the proceeding sentence for clarification.

Additional organizational issues: The results section of the paper includes much review of past literature. This is atypical and is confusing. I would suggest moving the review of past literature to the introduction or to the discussion, depending on the intended purpose of citing the research. This would also help with the organizational issues noted previously.

Author’s Response: For organization and clarity, we have move the review of past literature and discussion from the “Results and Discussion” section and added a stand-alone “Discussion” section to the manuscript (Pages 9-12).  

Some typos and word choice issues: There are missing periods and spaces and misplaced apostrophes throughout. Reporting non-white participants as “people of color” (p. 3) is vague. On page 4, a word is repeated: “who is responsible for responsible for the safety…” Typo on page 7: “land was used for hundred years ago…” On page 3, it is reported that “All interviewees (n = 17).” N in this case should be capitalized.

Author’s Response: We have thoroughly proof-read the article for typographical errors. We have modified the sentence to avoid vagueness (Page 3, lines 45-46).  We have also removed the repeated words “responsible for” (Page 5, line 46).

Conflicting information: On page 3, it’s reported that “Forty-seven percent of participants  described gardening as relaxing.” Several sentences later, it is reported that “Five participants cited gardening as being relaxing for a variety of reasons.” These two statements seem at odds with one another and lead me to question other reported results.

Author’s Response: We apologize for the confusion. We have modified the sentence for clarify (Page 4, line 9-10).

Definition of intergenerational learning: Intergenerational learning is defined as “knowledge passed from one generation, often parents or grandparents, down to the person in question” (p. 3). However, intergenerational learning also involves younger generations sharing knowledge and skills with older generations. This is made clear further on in the paragraph, but this particular sentence should be revised for clarity/accuracy.

Author’s Response: Thank you, you are correct. We have modified the definition of intergenerational learning to include younger generations sharing knowledge and skills with older generation (Page 4, lines 29-30).

Framing/word choice: Section 3.8 is labeled “study implications for environmental health” but the section feels mislabeled—it seems to discuss environmental implications for human health more than it does the implications of environmental health. Also in this section, what does it mean to “inform human health exposure assessments?” This language is unclear to me.

Author’s Response: Environmental health is a subsection of the field of Public Health and is defined as: “the study and management of environmental conditions that affect the health and well-being of humans” (An Introduction to Community and Public Health; McKenzie, Pinger, and Seabert, 2018).  “Inform human health exposure assessments” refers to exposure assessments that are conducted to estimate an individual’s exposure (dose) to a pollutant. Exposure assessments are when we look at how much of a pollutant is present in the environment and how someone might be exposed. This information is then used to calculate and estimate the exposure.

Figure 1: This figure isn’t particularly helpful and is even confusing. Shouldn’t all gardeners report watering, planting, and weeding? What are the other 13 gardeners doing that is considered gardening?

Author’s Response: This figure is reporting all the major activities and opportunities for direct contact with the soil. These responses were mostly derived from the prompt “What is your gardening routine?” This figure provides public health officials information regarding the various types of exposure routes for incidental soil ingestion for a gardener. Also, this further supports why when conducting an exposure assessment with communities, it is critical to determine whether they garden to then assume as soil ingestion as a potential exposure route to environmental contaminants of concern.   

Causal language error: Page 9 states that “Because gardening leads to an increased consumption of fruits and vegetables…” This is a causal statement that is not cited. To my knowledge, all research on gardening and fruit and vegetable consumption has been correlational. This language should be changed.

Author’s Response: The citation was added. Please refer to citations 30 and 31 (Page 11, line 28).

First sentence in conclusion: Related to the organizational issues already noted, this sentence is somewhat misleading. It does not accurately reflect what the article discusses. In particular, this sentence is the first place where “play and inquiry spaces for children” comes up, but the sentence states that the article demonstrates how gardens create these. Please revise this sentence to more accurately reflect what the paper discusses.

Author’s Response: Thank you. The first sentence of the conclusion (Page 12, lines 28-32) was modified.

The data reported in this article are rich! With some major revisions in organization and reporting of methodology, in addition to some careful editing and proofreading, I think this article could make an important contribution to the field.

Author’s Response: Thank you for your thorough and critical comments. By revising the organization of the manuscript, adding more details to the methodology section, and additional editing and proofreading, this manuscript has been substantially improved. 

Thank you for your time and consideration Reviewer 1!

Reviewer 2 Report

For me the novel aspect of the research lies in its focus on gardeners’ awareness of the soil and past uses of sites, which is an aspect of community gardening seldom addressed in research. The benefits associated with community gardening, especially the therapeutic benefits, are reasonably well documented elsewhere, with a great deal of literature on horticultural therapy already in existence. As such, this element of the paper does not provide new information. The authors have not stated how their work is novel or how it links to or develops the field.

My main concerns with the paper are surrounding its lack of theoretical engagement and the (at times) confusing structure. The biggest issue for me was the use of “intrinsic” and “extrinsic” categories of the benefits experienced by the participants. The authors mention Self-determination Theory but do not elaborate on the theory or develop it in any substantial way. They then categorise their results into intrinsic and extrinsic benefits but do not define extrinsic and intrinsic benefits or go into any details as to why the highlighted benefits fit into either category. Why is intergenerational learning intrinsic and community enhancement is extrinsic for example? Additionally, the relationship between motivations and benefits is not made clear. It is clear from the title and abstract that the research seeks to discuss benefits of community gardening rather than motivations but it seems that occasionally the line between the two can be blurred and the terms are conflated (for example on p2 line 9 and line 15). This may be helped by further elaboration of the theoretical framework used and how it can be developed to aid analysis of the results.

In general, I also feel that the reader would benefit from a more detailed explanation of the Boston Natural Areas Network to better understand the types of activities the network supports and how the participants are linked to the schemes.

Some more minor observations where some clarity would help the reader:

-        P2, line 1 – it is not clear if statistics are global/national etc. Is there any more recent evidence than 2013?

-        I am not sure if the title adequately reflects the content of the paper – I think the paper is about more than just intrinsic and extrinsic benefits and how these link to PH interventions.

-        Results section (p3, line 3) would benefit from a short introductory paragraph summarising the structure of the section and how the subsections have been divided and the rationale behind this.

-        P3, line 37 – it is not clear to me that community based environmental change or sustaining neighbourhoods are relevant in this section that focuses predominantly on mental health.

-        The section on intergenerational learning does not always make it clear that intergenerational learning is a benefit of community gardening (or a motivation) – if a participant has learnt to garden from their parents or grandparents, can this be seen as a benefit of community gardening? Would this be more of a motivation for becoming involved? (p4, paragraph 1).

-        P4, line 11-15 also discusses motivations rather than benefits of intergenerational activities

-         On p8, line 13, the paper makes reference to soil lead concentrations, “ranging from 305-1643 mg/kg”. It would be helpful if the authors provided context here – what is the recommended limit? When does it become dangerous?

I feel that the research has made a number of interesting observations and would benefit from more focused theoretical engagement and some efforts to improve structural clarity that can help guide the reader through the paper and emphasise key arguments.

Author Response

RESPONSE TO REVIEWER 2 COMMENTS

January 23, 2019

Dear Reviewer 2,

We are resubmitting an original manuscript: “Understanding the Intrinsic and Extrinsic Benefits and Challenges Associated with Community Gardening to Improve Environmental Public Health Prevention and Intervention” for publication in the International Journal of Environmental Research and Public Health. Thank you very much for the opportunity to revise and resubmit the manuscript.

We are grateful for the thoughtful comments we received from you regarding our manuscript. We appreciate the opportunity to revise the manuscript and submit it for your further consideration. We have taken into account all of your comments. Below are our responses to the comments.

-       For me the novel aspect of the research lies in its focus on gardeners’ awareness of the soil and past uses of sites, which is an aspect of community gardening seldom addressed in research. The benefits associated with community gardening, especially the therapeutic benefits, are reasonably well documented elsewhere, with a great deal of literature on horticultural therapy already in existence. As such, this element of the paper does not provide new information. The authors have not stated how their work is novel or how it links to or develops the field.

Author’s Response: Thank you for your helpful feedback. We have added the novel aspects of this work to the abstract (Page 1, lines 21-25), newly added discussion section (Pages 9-12), and conclusion (Page 12, lines 28-32).

-       My main concerns with the paper are surrounding its lack of theoretical engagement and the (at times) confusing structure. The biggest issue for me was the use of “intrinsic” and “extrinsic” categories of the benefits experienced by the participants. The authors mention Self-determination Theory but do not elaborate on the theory or develop it in any substantial way. They then categorise their results into intrinsic and extrinsic benefits but do not define extrinsic and intrinsic benefits or go into any details as to why the highlighted benefits fit into either category. Why is intergenerational learning intrinsic and community enhancement is extrinsic for example? Additionally, the relationship between motivations and benefits is not made clear. It is clear from the title and abstract that the research seeks to discuss benefits of community gardening rather than motivations but it seems that occasionally the line between the two can be blurred and the terms are conflated (for example on p2 line 9 and line 15). This may be helped by further elaboration of the theoretical framework used and how it can be developed to aid analysis of the results.

Author’s Response: Additional text defining extrinsic and intrinsic motivation had been added to the Introduction section (Page 2, lines 12-23). A rationale on how the highlighted outcomes were categorized is detailed in the methods under the sub-header “Coding” (Page 3). Lastly, the term “motivation” is properly used throughout the manuscript.

Intergenerational learning is considered intrinsic because it falls under a Self-Determination Theory sub-theory called “Cognitive Evaluation Theory” (elaborated upon on Page 10, lines 1-7). Intergenerational learning is a factor in social contexts that may produce variability in intrinsic motivation.

Community enhancement is extrinsic because it is a form of relatedness and falls under a Self-Determination Theory sub-theory called Organismic Integration Theory. Please refer to the added text in the Introduction (Page 2, lines 19-23) and new Discussion section (Page 10, line 49 through page 11, line 5).

-       In general, I also feel that the reader would benefit from a more detailed explanation of the Boston Natural Areas Network to better understand the types of activities the network supports and how the participants are linked to the schemes.

Author’s Response: We have added text about the Boston Natural Areas Network organization (Page 2, lines 35-37, and citation 14).

Some more minor observations where some clarity would help the reader:

-        P2, line 1 – it is not clear if statistics are global/national etc. Is there any more recent evidence than 2013?

Author’s Response: This is a statistic is for the United States. The sentence has been modified to read: “In 2013, 37 million American households…” (Page 1, line 39).

-        I am not sure if the title adequately reflects the content of the paper – I think the paper is about more than just intrinsic and extrinsic benefits and how these link to PH interventions.

Author’s Response: We have changed the title to: “Understanding the Intrinsic and Extrinsic Motivations Associated with Community Gardening to Improve Environmental Public Health Prevention and Intervention”.

-        Results section (p3, line 3) would benefit from a short introductory paragraph summarising the structure of the section and how the subsections have been divided and the rationale behind this.

Author’s Response: A short introductory paragraph summarizing the structure of the section and how the subsections have been divided and the rationale behind this has been added to the results section (Page 3, Lines 34-42).

-        P3, line 37 – it is not clear to me that community based environmental change or sustaining neighbourhoods are relevant in this section that focuses predominantly on mental health.

Author’s Response: The two sentences that referring to education and sustaining neighborhoods has been moved to Section 4 - Discussion (Page 10, lines 45-48).

-        The section on intergenerational learning does not always make it clear that intergenerational learning is a benefit of community gardening (or a motivation) – if a participant has learnt to garden from their parents or grandparents, can this be seen as a benefit of community gardening? Would this be more of a motivation for becoming involved? (p4, paragraph 1).

Author’s Response: Thank you for this comment. You are correct; learning to garden from your parents or grandparents is not necessarily a benefit. We agree and see intergenerational learning as an intrinsic form of motivation, not a benefit. Changes have been made, see Page 10-11.

-        P4, line 11-15 also discusses motivations rather than benefits of intergenerational activities

Author’s Response: Thank you. We have corrected the sentence (which has been modified and moved to Page 10).

-         On p8, line 13, the paper makes reference to soil lead concentrations, “ranging from 305-1643 mg/kg”. It would be helpful if the authors provided context here – what is the recommended limit? When does it become dangerous?

Author’s Response: The U.S. Environmental Protection Agency’s lead soil screening level and information from the U.S. Center for Disease Control about children’s blood lead levels was added, along with proper citation (Page 12, lines 17-26; citations 35 and 37).

-       I feel that the research has made a number of interesting observations and would benefit from more focused theoretical engagement and some efforts to improve structural clarity that can help guide the reader through the paper and emphasise key arguments.

Author’s Response: Thank you for your considerate comments. We have added further focus on our applied theoretical framework, improve the structure of the manuscript to help guide the reader, and emphasized our key arguments and literary contributions.

Thank you for your time and your consideration Reviewer 2.

Round  2

Reviewer 1 Report

The authors did a commendable job of revising the manuscript. Overall, the manuscript is much better organized, which makes for a clearer scientific contribution.

There is still one minor revision that I would like to request regarding a comment that was not fully addressed in my first review. On page 11, lines 28-29 and line 32 still use causal language when the language should be correlational. The studies cited show that there is an association between gardening and produce consumption, but not that gardening causes increased produce consumption. With few exceptions, only an experimental design with random assignment to conditions can show a causal relationship, and neither of the cited studies are experiments. Thus, I recommend the authors change their language appropriately (e.g., "Because gardening is associated with increased consumption of fruits and vegetables...") 

Author Response

RESPONSE TO REVIEWER 1 COMMENTS

February 1, 2019

Dear Reviewer 1,

We are grateful for the additional thoughtful comments we received from you regarding our manuscript. We appreciate the opportunity to address the highlighted minor revisions and submit it for further consideration. We have taken into account all of your comments and have highlighted the minor revisions in yellow (major revisions are still highlighted in blue). Below are our responses to the comments.

The authors did a commendable job of revising the manuscript. Overall, the manuscript is much better organized, which makes for a clearer scientific contribution.

There is still one minor revision that I would like to request regarding a comment that was not fully addressed in my first review. On page 11, lines 28-29 and line 32 still use causal language when the language should be correlational. The studies cited show that there is an association between gardening and produce consumption, but not that gardening causes increased produce consumption. With few exceptions, only an experimental design with random assignment to conditions can show a causal relationship, and neither of the cited studies are experiments. Thus, I recommend the authors change their language appropriately (e.g., "Because gardening is associated with increased consumption of fruits and vegetables...") 

Author’s Response: Thank you for your positive response and kind words. We are delighted to hear that the  manuscript is much better organized, which makes for a clearer scientific contribution.

Thank you for the additional clarification. We have changed the language to clearly state that there is an association between gardening and produce consumption, but not that gardening causes increased produce consumption. Please refer to page 11, lines 48 and 51 (changes are highlighted in yellow).

Thank you for your time and consideration Reviewer 1!

Reviewer 2 Report

The paper is much improved from my perspective. I've found the additional details on the theoretical framework (SDT) helpful in understanding how the motivations have been categorised. The structure and narrative is also much clearer, making the significance of the results more accessible from the reader's perspective. I have some very small suggestions for improvement.

 - A final proofread would be a good idea to make sure small typos are picked up (e.g. Abstract, p1, line 22, should read "lead" instead of "led"; p11, line 44 uses informal "who's" rather than who is (a stylistic preference perhaps))

 - p1, line 39 - I would like to see the sentence "Due to knowledge of these benefits, community gardening is on the rise" accompanied by a citation. I am not confident that this causality has been established but if it has, a reference would be helpful. If not, perhaps it should be reworded.

 - On p3, lines 18 and 34, the narrative slips into first person, with the use of "we". This is the only time I can see the use of this voice in the paper so it feels a little jarring/inconsistent.

 - p4, line 36 - I appreciate the authors' clarity on why intergenerational learning is an intrinsic form of motivation, however the quote used to evidence this does not make reference to the intergenerational aspect of learning. It simply states that the interviewee enjoyed learning from others. A quote that demonstrated this aspect would be more appropriate if the authors have evidence to this effect.

 - p11, lines 9-10 - the authors state that they have investigated "what type of governance model is needed to sustain urban agricultural endeavors like food gardening" but I am not convinced that they have shown this. Some additional commentary here would be helpful.

Author Response

RESPONSE TO REVIEWER 2 COMMENTS

February 1, 2019

Dear Reviewer 2,

We are grateful for the additional thoughtful comments we received from you regarding our manuscript. We appreciate the opportunity to address the highlighted minor revisions and submit it for further consideration. We have taken into account all of your comments and have highlighted the minor revisions in yellow (major revisions are still highlighted in blue). Below are our responses to the comments.

The paper is much improved from my perspective. I've found the additional details on the theoretical framework (SDT) helpful in understanding how the motivations have been categorised. The structure and narrative is also much clearer, making the significance of the results more accessible from the reader's perspective. I have some very small suggestions for improvement.

Author’s Response: Thank you for your positive response and kind words. We are delighted to hear that the  manuscript is much better organized and that the additional details on the theoretical framework (SDT) was helpful in understanding how the motivations have been categorized.

 - A final proofread would be a good idea to make sure small typos are picked up (e.g. Abstract, p1, line 22, should read "lead" instead of "led"; p11, line 44 uses informal "who's" rather than who is (a stylistic preference perhaps))

Author’s Response: Thank you. We have made the following corrections:

·      p1, line 22, reads "lead" instead of "led" (see yellow highlight)

·      p12, line 13 "who's" has been changed to “who is”

 - p1, line 39 - I would like to see the sentence "Due to knowledge of these benefits, community gardening is on the rise" accompanied by a citation. I am not confident that this causality has been established but if it has, a reference would be helpful. If not, perhaps it should be reworded.

Author’s Response: The sentence has been modified, additional details have been added, and it is now accompanied by a citation (page 1, lines 40-42).

 - On p3, lines 18 and 34, the narrative slips into first person, with the use of "we". This is the only time I can see the use of this voice in the paper so it feels a little jarring/inconsistent.

Author’s Response: We have corrected the sentence, please see page 3, line 19.

 - p4, line 36 - I appreciate the authors' clarity on why intergenerational learning is an intrinsic form of motivation, however the quote used to evidence this does not make reference to the intergenerational aspect of learning. It simply states that the interviewee enjoyed learning from others. A quote that demonstrated this aspect would be more appropriate if the authors have evidence to this effect.

Author’s Response: You are correct, thank you. We have added a two new quotes, please see page 4, lines 32-37. We have also moved the statement and quote you highlighted (and copied below) to section 3.3.2. Community enhancement and building relationships (page 5, lines 31-37).

Participants (N=5, 29%) also reported learning different techniques that were unique to their area from their neighbors.  One gardener stated they “…talked to all the Italian neighbors around me that used to have gardens and got their advice.” Below is a thought from a participant capturing their motivation for communal relationships:

When you’re gardening at home it’s one thing, but when you’re gardening with other people that are doing stuff that’s interesting, you can say how did you do that? Or why are mine dying and yours living?  It’s…really great…So you learn a lot from people definitely.

 - p11, lines 9-10 - the authors state that they have investigated "what type of governance model is needed to sustain urban agricultural endeavors like food gardening" but I am not convinced that they have shown this. Some additional commentary here would be helpful.

Author’s Response: The sentence has been modified, please refer to page 11, lines 28-29. Details are provided on the aforementioned theme in section 4.4 and a minor revision was made to reflect consideration of Reviewer 2’s comment (page 12, line 14).

Thank you for your time and consideration Reviewer2!
